# Distinct populations of cortical pyramidal neurons mediate drug reward and aversion

A. F. Garcia [1,2], E. A. Crummy [1,2], I. G. Webb[1], M. N. Nooney [1] & S. M. Ferguson [1,2,3✉]

Processing within the anterior cingulate cortex (ACC) is crucial for the patterning of appropriate behavior, and ACC dysfunction following chronic drug use is thought to play a major role in drug addiction. However, cortical pyramidal projection neurons can be subdivided into two major types (intratelencephalic (IT) and pyramidal tract (PT)), with distinct inputs and projection targets, molecular and receptor profiles, morphologies and electrophysiological properties. Yet, how each of these cell populations modulate behavior related to addiction is unknown. We demonstrate that PT neurons regulate the positive features of a drug experience whereas IT neurons regulate the negative features. These findings support a revised theory of cortical function in addiction, with distinct cells and circuits mediating reward and aversion.

[1] Center for Integrative Brain Research, Seattle Children's Research Institute, Seattle, WA, USA. [2] Neuroscience Graduate Program, University of Washington, Seattle, WA, USA. [3] Department of Psychiatry and Behavioral Sciences, Alcohol and Drug Abuse Institute, University of Washington, Seattle, WA, USA. ✉email: smfergus@uw.edu

Drug addiction is a prevalent and persistent disease marked biologically by molecular and synaptic adaptations in corticolimbic circuitry, and psychologically by a combination of positive and negative symptoms[1–4]. Accordingly, the misuse of drugs such as cocaine is driven by a complex interplay between the potent reinforcing effects of these drugs and the aversive states produced by extended and/or discontinued use, including stress and dysphoria[2–4]. The anterior cingulate cortex (ACC) is an intriguing, yet understudied, target for the dysregulated drug use characteristic of addiction as it responds to both positively and negatively valenced stimuli. For instance, the ACC is robustly activated by sexual mating and cocaine and -associated cues, as well as by chronic pain and remote fear memories[5–10]. In addition, the ACC plays an important role in decision-making, especially in tasks that require an evaluation of rewards and their associated costs[11]. The ACC is also notable anatomically, as it sends a strong glutamatergic project to the dorsomedial striatum (DMS), an area that is crucial for goal-directed actions and the transition to addiction[1,12].

To date, research examining the role of cortical neurons in addiction has not differentiated between the two major types of pyramidal projection neurons, which are physically intermingled but highly heterogeneous. Intratelencephalic (IT) cortical neurons project bilaterally to striatum and to contralateral cortex whereas pyramidal tract (PT) cortical neurons project much more diffusely, including ipsilaterally to striatum, the PT, and other deep structures including thalamus and the ventral tegmental area[13–15]. These neuronal populations also differ in their morphology, receptor and transcription factor expression, and electrophysiological properties[14,15]. Given the breadth and magnitude of biological differences between these two cell types, we hypothesized that IT and PT neurons would uniquely regulate behaviors related to cocaine addiction.

## Results

The inputs to IT and PT neurons have not been well-characterized in the rat, so we first examined them using a modified rabies system for monosynaptic retrograde tracing[16]. Cre-recombinase (Cre-)dependent viruses expressing an avian receptor to permit rabies infection (AAV5-EF1α-FLEX-TVA-mCherry) and a G-protein necessary for retrograde synaptic transport of rabies (AAV8-CA-FLEX-RG) were infused unilaterally into the ACC and a retrograde Cre virus (CAV2-Cre) was infused into the ipsilateral pyramidal tract or contralateral DMS to restrict AAV expression to PT or IT neurons, respectively. A modified rabies virus (EnvA G-deleted Rabies-eGFP) was then infused into the ACC, thus allowing for visualization of the direct projections to PT or IT neurons. We found that inputs to PT neurons were largely restricted to cortical neurons (Fig. 1e, 88% (22 of 25) of infected cells). In contrast, inputs to IT neurons were much more diverse, including an especially strong input from the thalamus (Fig. 1e, 36% (113 of 315) of infected cells), consistent with other CTB tracing studies[6]. The relatively limited input we observed to PT neurons may be a result of the rabies tracing method or due to species differences, as previous work in mice demonstrated that ACC PT neurons are also innervated by the thalamus[6].

To allow for transient, bilateral modulation of cell-specific neural activity, we used both a multi-recombinase-dependent Designer Receptor Exclusively Activated by Designer Drugs (DREADD) approach and an optogenetic approach. For selective expression in PT neurons, the inhibitory $hM_4Di$ receptor (AAV8-hSyn-DIO$^{LoxP}$-hM4Di-mCherry) or the stimulatory light-gated ion channel channelrhodopsin (AAV5-EF1a-DIO-hChR2-EYFP) was bilaterally infused into the ACC and CAV2-Cre was bilaterally infused into the pons. For selective expression in IT neurons, the DIO-$hM_4Di$ virus was infused into one ACC hemisphere and a retrograde AAV-expressing Cre (rAAV-Cre) was infused into the contralateral DMS. In addition, a Flppase (Flp-)dependent $hM_4Di$ virus (hSyn-FRT-hM4Di) was infused into the other ACC hemisphere and a retrograde AAv-expressing Flp (rAAV-Flp) was infused into the contralateral DMS (Fig. 1d). The outputs of PT and IT neurons were visualized using the mCherry tag of the DIO-$hM_4Di$ virus. Consistent with other reports[13], PT terminal expression was observed in striatum, thalamus, and other downstream structures previously identified whereas IT terminal expression was only evident in the striatum and cortex (Fig. 1c, d).

To determine if IT and PT neuronal populations in the rat ACC represent distinct, non-overlapping groups of cells, a retrograde cholera toxin subunit B (CTB) conjugated to AlexaFluor (AF) 488 (CTB-488) was infused unilaterally into the pons and a retrograde CTB conjugated to AF647 (CTB-647) was infused into the contralateral DMS (Fig. 2a). We found that 38% (33 of 86) of ACC neurons were labeled as putative PT (CTB-647+) neurons, 51.6% (45 of 86) were labeled as putative IT (CTB-488+) neurons, and 10.4% (8 of 86) were co-labeled with both fluorophores (Fig. 2a, b), suggesting that ~90% of IT and PT neurons comprise distinct neuronal populations, consistent with previous CTB tracing studies[6]. We next examined if this degree of segregation was sufficient to drive distinct behavioral responses by examining the effect of transiently decreasing IT and PT cortical activity on locomotor activity elicited by a compound stimulus (novel environment + cocaine). We found that the effect of inhibition significantly varied as a function of cell-type with IT inhibition increasing and PT inhibition decreasing activity (Fig. 2c, two-way RM ANOVA, group × time interaction: $F_{(29,667)} = 1.97$, $p < 0.002$). Together with the rabies and CTB tracing studies, these data provide evidence that IT and PT neurons can regulate behavioral activity through distinct cortical circuits.

The reinforcing/rewarding effects of cocaine can be measured using a conditioned place preference (CPP) paradigm, where animals that receive pairings of cocaine on one side of a chamber will spend more time on that side when allowed to freely explore the chamber[17] (Fig. 3a, b, paired $t$-test, $t_{(3)} = 5.49$, $p = 0.01$). We examined the effect of transiently decreasing IT and PT cortical activity on the development of a CPP to cocaine to test the hypothesis that the neuronal populations would have opposing modulatory effects on cocaine reward. Unexpectedly, we found that IT inhibition had no effect on a CPP for cocaine (Fig. 3c, paired $t$-test, $t_{(5)} = 2.31$, $p = 0.07$). However, animals spent significantly more time in the side of the chamber that had been paired with cocaine plus PT inhibition compared to the side of the chamber that had been paired with cocaine alone (Fig. 3d, paired $t$-test, $t_{(4)} = 7.18$, $p = 0.002$). In addition, pairing cocaine with optogenetic activation of PT neurons led to a significant decrease in the amount of time spent in that side of the chamber (Fig. 3e, paired $t$-test, $t_{(3)} = 8.00$, $p = 0.004$). The effects of PT neuronal modulation can be attributed to alterations in drug reward because neither inhibition (Fig. 3f, paired $t$-test, $t_{(8)} = 0.89$, $p = 0.40$) or activation (Fig. 3g, paired $t$-test, $t_{(3)} = 0.78$, $p = 0.49$) of PT neurons alone had an effect on chamber preference nor did it modulate a hedonic food reward (Fig. 3i, paired $t$-test, $t_{(5)} = 0.32$, $p = 0.76$).

The aversive properties of cocaine can be assessed using a conditioned taste avoidance (CTA) paradigm (Fig. 4a), where administration of high doses of cocaine following consumption of a sucrose solution will result in subsequent avoidance of the sucrose[18,19] (Fig. 4b–d: two-way RM ANOVA, group × session interaction: $F_{(3,45)} = 12.62$-$22.21$, $p < 0.0001$; Sidak's, $p < 0.0001$, cocaine vs. saline, sessions 2–4; Fig. 4e–g: unpaired $t$-test,

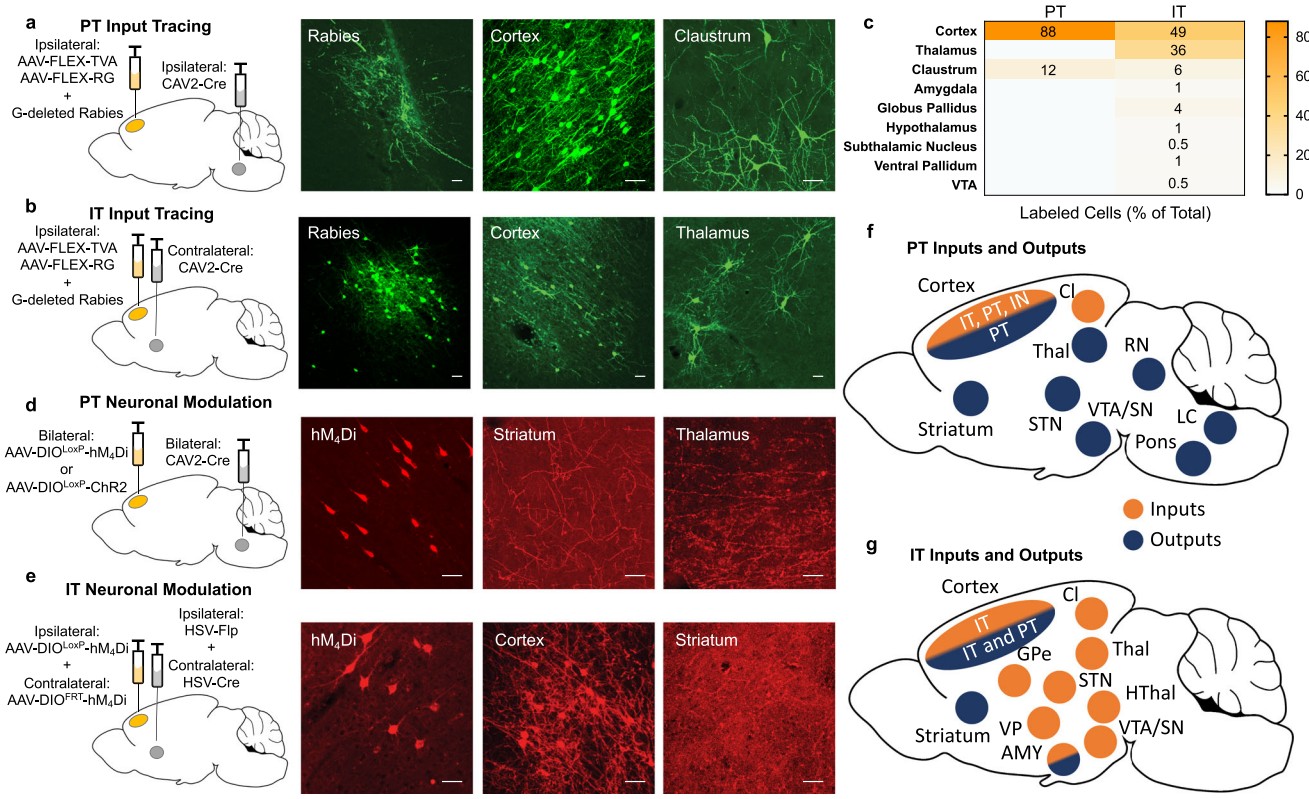

**Fig. 1 PT and IT neurons in ACC have distinct inputs and outputs. a, b** Viral strategy using a modified rabies system for monosynaptic retrograde tracing in PT (**a**) or IT (**b**) neurons. Retrograde CAV2-Cre virus was injected unilaterally into the pyramidal tract (**a**) or the dorsomedial striatum (IT) and Cre-dependent TVA and RG vectors were infused into the ipsilateral ACC along with a modified EnvA G-deleted rabies-eGFP. Representative images of GFP-tagged rabies immunofluorescence in PT (**a**, left panel) and IT (**b**, left panel) neurons at the site of infection in the cortex, PT inputs from cortex and claustrum (**a**, middle and right panels), and IT inputs from cortex and thalamus (**b**, middle and right panels). These regions represent the two strongest inputs into PT (**a**) and IT (**b**) cells. Scale bars, 50 μm. **c** Summary of the relative input strength (by %) to PT and IT neurons in the ACC. Data from 25 PT cells and 315 IT cells. **d, e** Viral strategy for targeting transgenes bilaterally to PT (**d**) or IT (**e**) neurons. For PT neurons (**d**), CAV2-Cre was injected bilaterally into the pyramidal tract and a Cre-dependent hM$_4$D$_i$ or ChR2 virus was injected bilaterally into the ACC. For IT neurons (**e**), retrograde HSV-Cre and HSV-Flp viruses were injected unilaterally into opposite hemispheres of dorsomedial striatum and the complementary Cre-dependent or Flp-dependent hM$_4$D$_i$ virus was infused unilaterally into the contralateral ACC. Representative images of mCherry-tagged hM$_4$D$_i$ immunofluorescence in PT (**d**, left panel) or IT (**e**, left panel) neurons in cortex, PT terminals in striatum and thalamus (**d**, middle and right panels), and IT terminals in cortex and striatum (**e**, middle and left panels). The cortex image in **e** (middle panel) also shows hM$_4$D$_i$-labeled IT cell bodies because of contralateral IT projections. Scale bars, 50 μm. **f, g** Summary of the known PT (**f**) and IT (**g**) inputs and outputs. PT neurons have diverse outputs, but a restricted source of inputs originating in cortex and claustrum whereas IT neurons have a restricted set of outputs to contralateral cortex, striatum, and amygdala, but a diverse set of inputs.

$t_{(15)} = 6.28$–$7.60$, $p < 0.0001$). We examined the effect of transiently decreasing IT and PT cortical activity on the development of a CTA to test the hypothesis that the neuronal populations would differentially regulate the aversive effects of cocaine. We found that although PT neuron inhibition had no effect on cocaine-induced avoidance of sucrose consumption (Fig. 4b–d: two-way RM ANOVA, group × session interaction: $F_{(6,81)} = 1.1$–$3.5$, $p = 0.006$–$0.37$; Sidak's, $p > 0.05$ PT hM$_4$Di vs. control; Fig. 4e–g: one-way ANOVA, main effect of treatment: $F_{(2,26)} = 1.76$–$10.28$, $p = 0.0005$–$0.37$; $p > 0.05$ PT hM$_4$Di vs. control), inhibition of IT neurons significantly attenuated the reduction in sucrose consumption normally seen following cocaine administration (Fig. 4b: two-way RM ANOVA, group × session interaction: $F_{(6,81)} = 2.5$, $p = 0.03$, Sidak's, $p = 0.006$, IT hM$_4$Di vs. control, session 2; Fig. 4e: one-way ANOVA, main effect of group: $F_{(2,26)} = 6.20$, $p = 0.006$; Dunnett's, $p = 0.008$, IT hM$_4$Di vs. control). Interestingly, the amount of time spent drinking per bout was similar across groups (Fig. 4d: two-way RM ANOVA, group × session interaction not significant: $F_{(6,81)} = 1.10$, $p = 0.37$; Fig. 4g: one-way ANOVA, main effect of treatment: $F_{(2,26)} = 1.76$,

$p = 0.19$; Dunnett's, $p = 0.19$, IT hM$_4$Di vs. control); however, animals that received IT inhibition had significantly more drinking bouts (Fig. 4d: two-way RM ANOVA, group × session interaction; $F_{(6,81)} = 3.53$, $p = 0.004$; Sidak's, $p = 0.0001$, IT hM$_4$Di vs. control, session 2; Fig. 4g: one-way ANOVA, main effect of group: $F_{(2,26)} = 10.28$, $p = 0.0005$; Dunnett's, $p = 0.0004$, IT hM$_4$Di vs. control), indicating that inhibition of these neurons blocked cocaine's ability to reduce approach behavior towards the sucrose solution. In addition, CNO had no effect on sucrose consumption (Fig. 4h–j, two-way RM ANOVA, group × session interaction; $F_{(3,18)} = 0.18$–$0.76$, $p = 0.53$–$0.91$) or the development of a cocaine-induced CTA (Fig. 4k–m, two-way RM ANOVA, group × session interaction; $F_{(3,21)} = 0.83$–$2.18$, $p = 0.12$-$0.83$) in non-DREADD controls, and IT inhibition had no effect on sucrose consumption (Fig. 4n–p, two-way RM ANOVA, group × session interaction; $F_{(3,39)} = 1.28$–$1.49$, $p = 0.23$–$0.30$). Together, these data lend support to the notion that the CTA paradigm does indeed reflect a negative component of the drug experience, as a manipulation that greatly enhanced cocaine reward (PT inhibition) had no effect on a cocaine-induced CTA

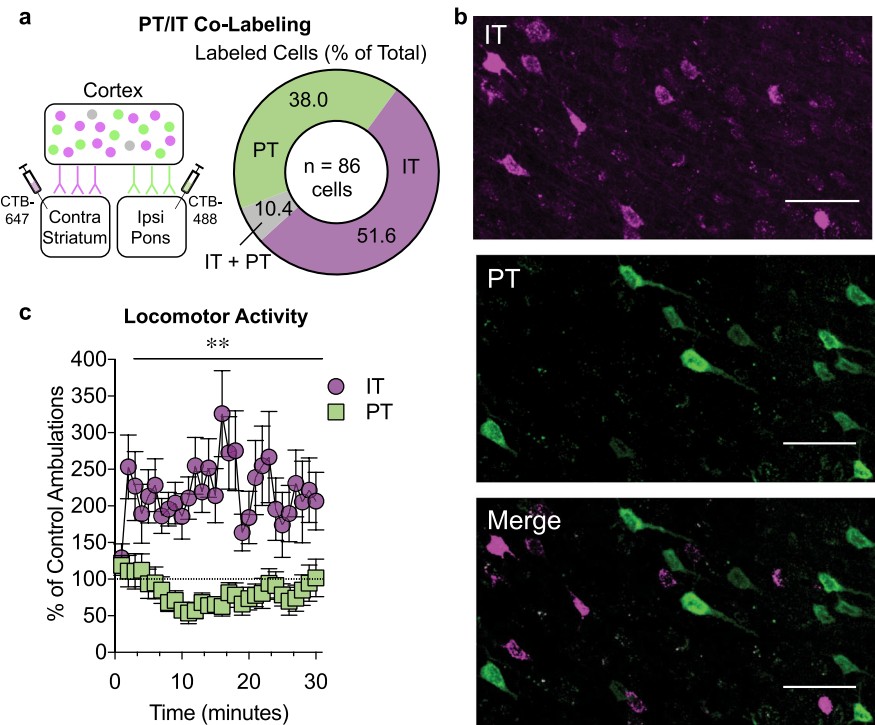

**Fig. 2 PT and IT neurons in ACC are anatomically and functionally distinct. a** Left: Viral strategy for anatomical tracing of IT and PT neurons with CTB ($N = 3$). Right: Quantification of neuronal labeling. Thirty-eight percent of labeled cortical neurons are PT+ (green), ~52% are IT+ (magenta), and ~10% are co-labeled from 86 total cells counted. **b** Representative photomicrographs of neuronal labeling of IT cells (top), PT cells (middle), and a merged image (bottom). Scale bar = 50 μm. **c** Line graphs where data points represent the mean ± SEM percent of control (vehicle DREADD, $N = 11$–12) ambulations induced by a compound stimulus (novelty + 15 mg/kg cocaine, i.p.). IT inhibition (magenta, $N = 14$) significantly increased and PT inhibition (green, $N = 11$) significantly decreased the percent of control activity, indicating that the cell types are functionally distinct. **$p < 0.001$, group × time interaction, two-way RM ANOVA.

whereas a manipulation that did not alter the rewarding properties of cocaine (IT inhibition) reduced the cocaine-induced CTA. This is in contrast to the idea that a cocaine CTA is driven by cocaine's strength as a reinforcer occluding the rewarding aspects of the sucrose[19]. It is also worth noting that IT inhibition was most effective during the first cocaine pairing, suggesting that IT neurons are important for modulating the initial aversive effects of cocaine. However, this finding raises the possibility that other systems (e.g. hindbrain areas) that can regulate the development of a CTA come on-line with more conditioning, leading to a reduced role of IT neurons.

## Discussion

In conclusion, we have identified that distinct sets of cortical neurons (PT and IT) regulate the positive and negative components of a drug experience. This research provides the first evidence for the differential role of these cell populations in the reinforcing and aversive properties of cocaine, and demonstrates that separate circuits (both anatomically and functionally) exist among prefrontal cortex pyramidal neurons. Interestingly, studies in pain models of preference and aversion show that PT stimulation and inhibition have opposite effects on preference and aversion, which were input-dependent[6], suggesting that the results observed here are unique to drug administration contexts. Prefrontal cortex hypoactivity has long been considered a dangerous consequence of chronic drug use, as it results in a loss of executive control and an increase in impulsivity[20,21]. Our data provide important new insights into the scope of this hypoactivity, as it suggests that a hypoactive prefrontal cortex would enhance the rewarding components of drug use (through a loss of

PT activity) while simultaneously reducing the negative components (through a loss of IT activity). Accordingly, these events would produce a significant shift in reward evaluation that favors continued drug use. This evidence of divergent roles for subsets of cortical pyramidal projection neurons in the positive and negative features of drug use, therefore, has fundamental implications for our understanding of cortical function in decision-making processes, both in drug addiction and more broadly.

## Methods

**Animal use.** All experiments were approved by the Seattle Children's Research Institute Institutional Animal Use and Care Committee and adhered to National Institutes of Health guidelines. Male Sprague Dawley rats (Envigo) weighing 250–274 g upon arrival acclimatized to the environment for at least 3 days prior to any experimental manipulation. All rats were pair housed for the duration of the experiments. The housing environment was maintained on a 12 h light/dark cycle and contained temperature and humidity control. Food and water were available ad libitum.

**Drugs.** Clozapine-N-oxide (CNO) was obtained from the NIH as part of the Rapid Access to Investigate Drug Program funded by the NINDs. CNO was administered into the intraperitoneal cavity (i.p.) in a volume of 1 mL/kg at a dose of 5 mg/kg. CNO was first dissolved in 100% dimethyl sulfoxide (DMSO), and then diluted in sterile water for a final concentration of 5% DMSO. Vehicle injections were 5% DMSO in sterile water. Cocaine HCl (obtained from National Institute on Drug Abuse) was dissolved in sterile 0.9% saline and administered in a volume of 1 mL/kg at doses of 10 mg/kg, i.p. (cocaine CPP) or 30 mg/kg, s.c. (sucrose CTA).

**Viral vectors.** A Cre-dependent hM$_4$Di-DREADD (AAV8-hSyn-DIO-hM$_4$Di-mCherry) viral vector was developed by Bryan Roth (University of North Carolina) and packaged in adeno-associated virus (AAV, serotype 8) at Addgene with an approximate titer of $1 \times 10^9$ viral genomes per μL. A Flp-dependent hM$_4$Di-DREADD (AAV1-hSyn-FRT-hM$_4$Di-YFP) was obtained from Larry Zweifel (University of Washington). A canine adenovirus-expressing Cre-recombinase

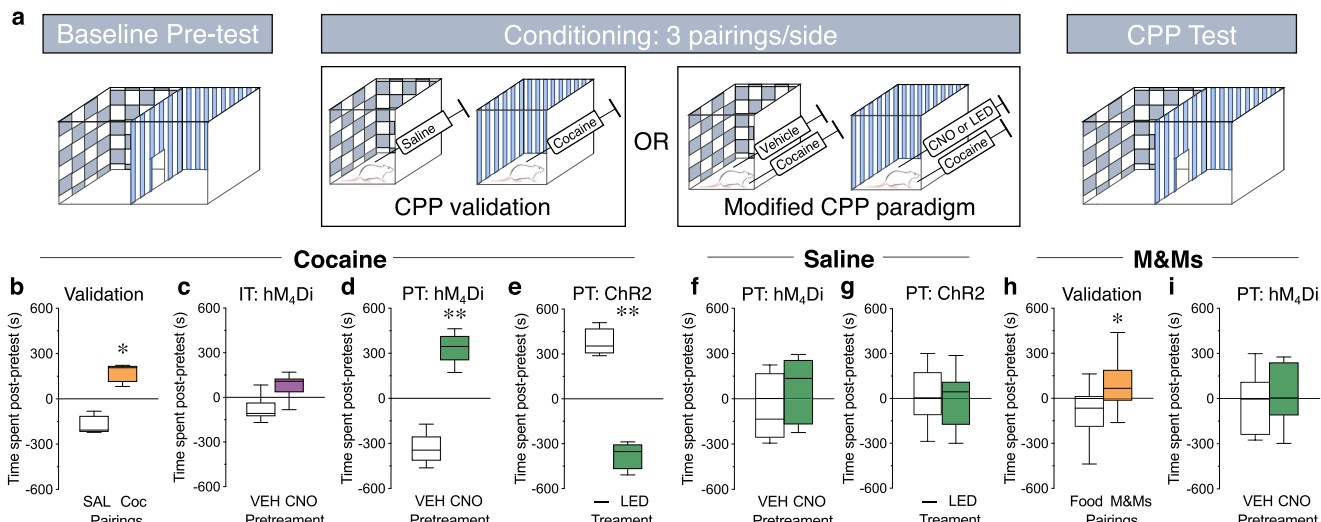

**Fig. 3 PT inhibition enhances the rewarding properties of cocaine. a** Baseline: Rats freely explore chamber for 20 min. Conditioning: Rats receive three pairings/side of saline and 10 mg/kg cocaine, i.p. (for conditioned place preference (CPP) validation) or vehicle (VEH) + cocaine and CNO (5 mg/kg, i.p.) (or LED light; 20 Hz, 20 ms pulses) + cocaine (modified CPP paradigm) over 3 days (2 conditioning sessions/day). CPP test: 24 h post-conditioning, rats freely explore chamber for 20 min. **b–i** Bar and whisker plots where bars represent the mean ± SEM difference in time spent on a side between the post-conditioning CPP test and the Baseline pretest. **b** Validation: Rats spent significantly more time on the cocaine-paired side (orange) than the saline-paired side, indicating a CPP to cocaine. N = 4. **c** IT hM$_4$Di: There was no difference in the time spent on the CNO + cocaine-paired side (purple) compared to the VEH + cocaine-paired side, indicating that IT inhibition does not modulate cocaine reward. N = 6. **d** PT hM$_4$Di: Rats spent significantly more time on the CNO + cocaine-paired side (green) than the VEH + cocaine-paired side, indicating that PT inhibition enhances cocaine reward. N = 6. **e** PT ChR2: Rats spent significantly less time on the light + cocaine-paired side (green) compared to cocaine alone, indicating that PT stimulation decreases cocaine reward. N = 4. **f** There was no difference in the time spent when rats received VEH or CNO (green) prior to saline on each side, indicating that PT inhibition alone is not reinforcing. N = 5. **g** There was no difference in the time spent when rats received light + saline (green) compared to saline alone, indicating that PT stimulation alone is not aversive. N = 4. **h** Validation: Rats spent significantly more time on the M&M candy-paired side (orange) compared to the food-paired side, indicating a CPP to a hedonic natural reward. N = 9. **i** PT hM$_4$Di: There was no difference in the time spent on the CNO + M&M-paired side (green) compared to the VEH + M&M-paired side, indicating that PT inhibition does not modulate the rewarding properties of a natural reinforcer. N = 6, *p < 0.05, **p < 0.01, paired t-test.

(CAV2-Cre) had a titer of approximately $2.5 \times 10^9$ viral genomes per mL and was prepared as previously described[22] and obtained from John Neumaier (University of Washington). EnvA G-deleted rabies vector (EnvA G-deleted Rabies-eGFP) was developed by Ed Callaway (Salk Institute) and obtained from the Salk Institute Viral Vector Core with a titer of approximately $2.91 \times 10^7$. The Cre-dependent rabies glycoprotein (AAV8-CA-FLEX-RG) and the Cre-dependent TVA receptor (AAV5-EF1a-FLEX-TVAmCherry) were obtained from the University of North Carolina viral vector core with an approximate titer of $1 \times 10^{12}$. The Cre-dependent channelrhodopsin (AAV5-EF1a-DIO-hChR2(H134R)-EYFP) was obtained from the University of North Carolina. The HSV-Cre (hEF1α-EYFP-IRES-Cre) and HSV-Flp (hEF1α-EYFP-IRES-Flpo) viral vectors were obtained from Dr. Rachael Neve (Harvard) at an approximate titer of $3.5 \times 10^9$.

**Viral targeting strategies.** Viral strategies for targeting transgenes selectively to PT and IT neurons were developed based on the unique output targets of the two cell populations. Specifically, PT neurons project ipsilaterally and send their primary axon to the pons. Although both PT and IT neurons project ipsilaterally to striatum, only IT neurons send contralateral projections to striatum[12,14]. Thus, if a recombinase-dependent viral vector is infused into one hemisphere of ACC, then infusing a retrograde recombinase viral vector into the ipsilateral pons will result in transgene expression selectively in PT neurons whereas infusing a retrograde recombinase viral vector into the contralateral striatum will result in transgene expression selectively in IT neurons. Although this unilateral approach is sufficient for anatomical studies, a bilateral targeting approach is necessary for behavioral studies in order to achieve neuronal modulation in both hemispheres. For PT neurons, this can be achieved by infusing a recombinase-dependent viral vector bilaterally into the ACC and a retrograde recombinase viral vector bilaterally into the pons. For IT neurons, this can be achieved by infusing different recombinase-dependent viral vectors unilaterally into the two ACC hemispheres and infusing the corresponding retrograde recombinase viral vectors unilaterally into the contralateral striatum.

**Surgical techniques.** During all surgical procedures, rats were anesthetized with isoflurane (4% induction, 2% maintenance, inhalation) and received meloxicam (1 mg/kg, i.p.) prior to surgery for analgesia. Using standard stereotaxic procedures,

33-gauge needles attached to gas-tight Hamilton syringes were placed above the region of interest. The following stereotaxic coordinates relative to Bregma (in mm) were used for virus injections (presented as brain region, anterior/posterior, medial/lateral, dorsal/ventral, injection volume in μL): ACC, +2.5, ±0.7, −1.9, 0.5; DMS, +0.2, ±2.0, −4.1, 1.0; pyramidal tract, −9.2, ±0.4, −9.8, 0.5. All viruses were infused into a region of interest at a rate of 400 nL/min, and needles were left in place for an additional 5 min to allow for diffusion away from the injection site. For targeting PT neurons, animals received CAV2-Cre infusions into the pyramidal tract (0.5 μL/side) and a Cre-dependent virus in ACC (chemogenetic experiments: AAV8-hSyn-DIO-hM4Di-mCherry, 0.5 μL/side; optogenetic experiments: AAV5-EF1a-DIO-hChR2(H134R)-EYFP, 0.5 μL/side; rabies tracing experiments: AAV8-CA-FLEX-RG (0.3 μL/side), AAV5-EF1a-FLEX-TVAmCherry (0.3 μL/side), and EnvA G-deleted Rabies-eGFP (0.5 μL/side)). Infusions were unilateral for rabies tracing and bilateral for chemogenetic and optogenetic experiments. For unilateral targeting of IT neurons for rabies tracing, animals received CAV2-Cre infusions unilaterally into DMS (1.0 μL) and a Cre-dependent virus in the contralateral side of ACC (AAV8-CA-FLEX-RG (0.3 μL), AAV5-EF1a-FLEX-TVAmCherry (0.3 μL), and EnvA G-deleted Rabies-eGFP (0.5 μL). For bilateral targeting of IT neurons, for chemogenetic experiments, animals received unilateral infusions of hEF1α-EYFP-IRES-Cre (1.0 μL) and hEF1α-EYFP-IRES-Flpo (1.0 μL) into opposite hemispheres of DMS and the complementary Cre or Flp-dependent vector into the contralateral side of ACC (AAV1-hSyn-FRT-hM4Di-YFP or AAV8-hSyn-DIO-hM4Di-mCherry, 0.5 μL). Rats had at least 3 days of post-operative recovery and monitoring following stereotaxic infusions.

**Rabies tracing.** Rats (n = 4) received infusions of helper viruses (AAV8-CA-FLEX-RG and AAV5-EF1a-FLEX-TVAmCherry) into ACC and CAV2-Cre either unilaterally into DMS (IT-targeting) or pons (PT-targeting) as described above. After 21 days, the animals received unilateral infusions of EnvA G-deleted Rabies-eGFP into ACC. The rabies virus was allowed 7 days to express before the rats were anesthetized with Beuthanasia-D (Patterson Veterinary) and transcardially perfused with phosphate-buffered saline (PBS) followed by 4% paraformaldehyde (PFA) in PBS. Control animals underwent the same procedures, but did not receive the CAV2-Cre injections. Brains were removed, post-fixed overnight, and switched to PBS the next morning. Brains were sectioned in 40 μM slices using a Leica vibrating microtome. Images across the rostro-caudal axis of the brains were taken

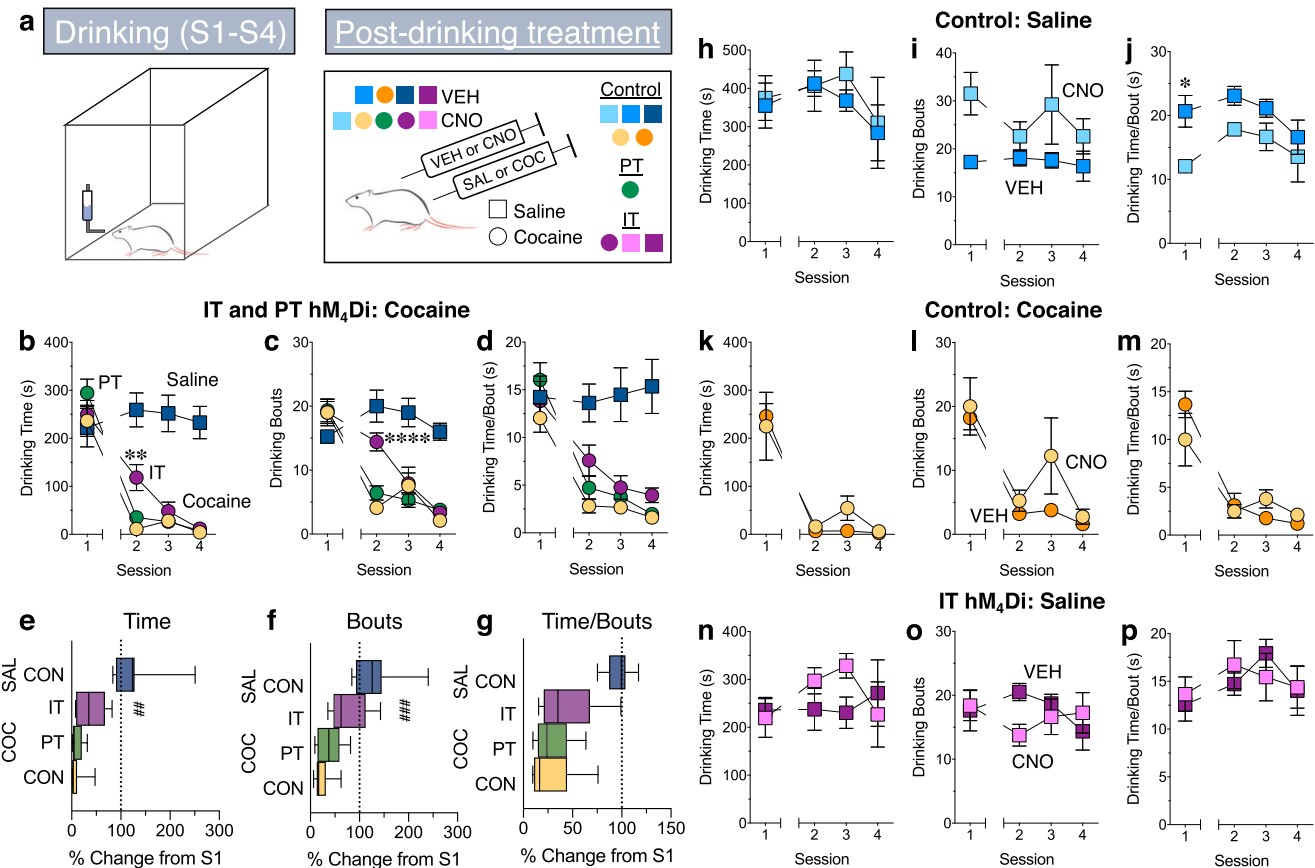

**Fig. 4 IT inhibition attenuates the aversive properties of cocaine. a** Rats are given access to a 15% sucrose solution for four 30 min sessions. VEH or CNO (5 mg/kg, i.p.) are administered after each session followed 15 min later by saline or cocaine (30 mg/kg, s.c.). Session 1 represents baseline sucrose drinking. **b–d**, **h–p** Line graphs of mean ± SEM total drinking time (**b**, **h**, **k**, **n**), number of drinking bouts (**c**, **i**, **l**, **o**) and amount of time spent drinking per bout (**d**, **j**, **m**, **p**). **e–g** Bar and whisker plots of mean ± SEM percent change from Session 1 to Session 2. **b–g** Post-drinking saline administration (dark blue, $N = 8$) had no effect, whereas post-drinking cocaine administration (light orange, $N = 9$) decreased sucrose consumption in sessions 2–4, indicating a sucrose CTA. **b**, **c** $p < 0.0001$, two-way RM ANOVA, Sidak's post hoc test, saline vs. cocaine groups. **e–g** $p < 0.0001$, unpaired $t$-test, saline vs. cocaine groups. PT inhibition (green, $N = 9$) had no effect whereas IT inhibition (purple, $N = 11$) attenuated the reduction in sucrose drinking time and the number of drinking bouts during Session 2. There were no group differences in the amount of drinking time per bout, indicating that IT inhibition blunted the ability of cocaine to reduce approach behavior. **b**, **c** $**p = 0.006$, $****p = 0.0001$, two-way RM ANOVA, Sidak's post hoc test, cocaine groups. **e–g** $##p = 0.008$, $###p = 0.0004$, one-way ANOVA, Dunnett's test, cocaine groups. **h–j** Although baseline consumption was different, CNO (light blue) had no effect compared to VEH (dark blue) in non-DREADD rats given sucrose, $N = 4$/group, $*p < 0.05$, two-way RM ANOVA, Sidak's post hoc test, VEH vs. CNO groups. **k–m** CNO (light orange) had no effect compared to VEH (dark orange) on a cocaine-induced sucrose CTA in non-DREADD rats, $N = 4$–5/group.
**n–p** IT inhibition (light purple, $N = 8$) had no effect on total drinking time, number of drinking bouts or amount of time spent drinking per bout compared to control (dark purple, hM$_4$Di IT rats receiving VEH, $N = 7$), indicating that the effects of IT inhibition on a cocaine-induced sucrose CTA were not due to a general reduction in sucrose consumption.

with confocal microscopy (×20; Zeiss LSM 710). eGFP-positive cells were quantified in ImageJ (v1.52s; NIH) and were assigned to brain structures based on standard anatomical landmarks using the Rat Brain Atlas.

**CTB tracing**. Rats ($n = 3$) received an infusion of CTB conjugated to AlexaFluor 488 (CTB-488; C-34775, Thermo Fisher) unilaterally into the pons (350 nL, 70 nL/min), and an infusion of CTB conjugated to AlexaFluor 647 (CTB-647, C-34778, Thermo Fisher) unilaterally into the contralateral DMS (350 nL/140 nL/ min). Twenty-one days later, rats were anesthetized with Beuthanasia-D (Patterson Veterinary) and transcardially perfused with PBS, followed by 4% PFA. Brains were extracted, fixed overnight in 4% PFA, and post-fixed in 30% sucrose. Brains were sectioned at 40 μm with a vibrating microtome. Z-stacks of the rostro-caudal axis of the ACC were collected with confocal microscopy (×20; Zeiss LSM 710) and localization of CTB-488 and CTB-647 were quantified using ImageJ (v1.52s; NIH).

**Cocaine-induced locomotor activity**. Rats ($N = 53$) were stereotaxically infused with DREADDs (hM$_4$Di) targeting either PT neurons or IT neurons as describe above and were given a minimum of 14 days to allow for viral expression prior to undergoing behavioral sessions. To assess locomotor activity following transient inhibition of these cell types, rats received pretreatment injections of either

5 mg/kg, i.p. CNO or vehicle 30 min prior to an injection of cocaine (15 mg/kg, i. p.). Rats were placed into locomotor activity boxes (San Diego Instruments) and allowed to habituate in the locomotor boxes 30 min prior to pretreatment. Ambulations were recorded in 1-min intervals, and defined as two consecutive infrared beam breaks over 30 min. A total of five rats were excluded from analysis due to lack of viral expression, resulting in the final groups: IT vehicle ($n = 11$), IT CNO ($n = 14$), PT vehicle ($n = 12$), PT CNO ($n = 11$).

**Conditioned place preference**. Rats ($N = 44$) underwent CPP testing using a two-chamber apparatus ($24 \times 12 \times 12$ in, with a divider in the middle). The two chambers differed in their wall patterns (horizontal vs. vertical stripes). Animals were first allowed to freely explore the apparatus for 20 min. Over the following 3 days, animals underwent six 20 min conditioning sessions (one in the morning and one in the afternoon separated by at least 4 h). During these conditioning sessions, the rats were administered a treatment and restricted to one side of the chamber. The order that they received the treatments was counterbalanced each day. The day after the last set of conditioning sessions, animals were allowed to freely explore the chamber for 20 min. Time spent in each chamber during pre- and post-conditioning test sessions were scored from videos (collected on a ELP USB camera and viewed on a VLC media player) by an experimenter blind to the conditions.

For cocaine experiments, rats received 10 mg/kg cocaine (i.p.) immediately prior to being placed into the chamber for experiments designed to produce a CPP. For chemogenetic experiments, rats received CNO (5 mg/kg, i.p.) 30 min before cocaine injections and placement into one side of the conditioning chamber and VEH 30 min before cocaine injections and placement in the other side. For optogenetic experiments, animals received stimulation via head-mounted blue LED with parameters of 20 Hz, 20 ms pulse duration, for 60 s with 30 s of no stimulation. These parameters have been shown to induce neuronal activation without cellular toxicity. Animals received stimulation in ACC (for stimulation of PT cell bodies). For food experiments, animals were placed into a chamber containing four M&Ms. Animals were not food deprived as we intended to test their response to a naturally hedonic reward as opposed to the alleviation of a negative state (hunger). Animals were previously exposed to M&Ms 2 days prior to behavioral testing to overcome neophobic suppression of eating. In order to test if cortical modulation specifically affected the rewarding aspects of cocaine or a hedonic food reward, animals received the rewarding stimulus (cocaine or M&Ms) on both sides of the CPP chamber. However, they only received CNO or optogenetic stimulation on one side of the chamber. This modified CPP approach produces a direct comparison and choice between the reward and cortical modulation plus reward and allows us to determine how cortical manipulation affects the relative value of the presented rewards.

**Conditioned taste aversion**. Rats ($N = 47$) underwent conditioned taste aversion (CTA) testing. One day before testing began, animals were allowed to explore and habituate to the cage for 30 min. During testing, animals were placed into a cage with access to 15% sucrose solution. At the end of the session, animals received either vehicle (5% DMSO/95% sterile water, i.p.) or CNO (5 mg/kg, CNO, i.p.) injections followed 15 min later by either saline (0.9%, s.c.) or cocaine (30 mg/kg, s.c.) injections. Animals underwent four 30 min conditioning sessions across 7 days. Time spent drinking was scored from videos (collected on a ELP USB camera and viewed on a VLC media player) by an experimenter blind to the conditions.

**Virus localization**. Rats were anesthetized with Beuthanasia-D and transcardially perfused with PBS followed by 4% PFA. Brains were removed, post-fixed overnight, and switched to PBS the next morning. Brains were sectioned in 40 μM slices using a Leica vibrating microtome. Floating sections were washed in 0.2% Triton-X/PBS solution for 10 min and blocked in 0.2% Triton-X/PBS solution/5% normal goat serum for 30 min at room temperature. Sections were incubated in 0.2% Triton-X/PBS solution/5% normal goat serum/primary antibody against dsRed (1:400, rabbit host, Clontech, #632496) or GFP (1:400, mouse host, Millipore, MAB3580) while on a standard analog shaker (VWR) overnight at room temperature. Sections were rinsed three times in 0.2% Triton-X/PBS solution for 10 min. They were blocked in 0.2% Triton-X/PBS solution/5% normal goat serum for 30 min at room temperature. Then they were incubated in 0.2% Triton-X/PBS solution/5% normal goat serum/goat anti-rabbit Alexa568-conjugated secondary antibody (1:250, Invitrogen, A-11036) or goat anti-mouse Alexa488-conjugated secondary antibody (1:250, Invitrogen, A-11029) for 2 h at room temperature on a standard analog shaker (VWR). Sections were rinsed in 0.2% Triton-X/PBS solution for 10 min followed by PBS for 10 min, mounted on slides, and coverslipped with Vectashield containing DAPI mounting medium (Vector Labs, H-1500). Z-stacks were captured using a Zeiss LSM 710 confocal microscope, and images were processed using ImageJ software (NIH).

**Data analysis**. All statistical analyses were determined a priori. Data were imported to Microsoft Excel 12 and GraphPad Prism 8 was used for statistical analyses. For the locomotor activity experiment, ambulations for CNO groups were normalized as percent of vehicle ambulations and data were analyzed across the session for the IT and PT groups using two-way repeated-measures analysis of variance (ANOVA) (group × time). For the CPP experiments, the effects of treatment (pre- vs. post-test) on time spent in each chamber were analyzed using a paired $t$-test. For the CTA experiments, the effects of treatment and conditioning on time spent drinking, drinking bouts, and time spent drinking per bout were analyzed using two-way repeated-measures ANOVA (group × session) followed by Sidak's post hoc test. The percent change from Session 1 to Session 2 on these measures were analyzed using unpaired $t$-tests (saline vs. cocaine) or one-way ANOVA followed by Dunnet's post hoc test (cocaine groups). For all comparisons, $\alpha \leq 0.05$.

**Reporting summary**. Further information on research design is available in the Nature Research Reporting Summary linked to this article.

## Data availability
Source data are provided with this paper.

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

## Acknowledgements
We thank Michelle Seo for assisting in behavioral experiments. This work was supported by grants from the National Institute on Drug Abuse (T32DA007278 to G.A.F. and R01DA036582 to F.S.M.) and from the National Science Foundation (DGE-1762114 to C.E.A.).

## Author contributions
G.A.F., C.E.A. and W.I.G. performed the behavioral experiments. G.A.F., C.E.A. and N.M.N. performed data analysis. G.A.F., C.E.A. and F.S.M. designed the experiments and wrote the manuscript.

## Competing interests
The authors declare no competing interests.

**Additional information**

