## [Peer Review File · Nature Communications]

Reviewers' Comments:

Reviewer #1:

Remarks to the Author:

This is a provocative, exciting study that has potential to shift the field's thinking on the neural substrates underlying the rewarding and aversive effects of drugs of abuse. The rationale for the study and the significance of the findings are clearly articulated in most cases. The methods, in terms of targeting the IT/PT neuron populations are state-of-the-art, and the behaviors under study are well chosen. The manuscript has no significant weaknesses.

Reviewer #3:

Remarks to the Author:

The rebuttal document includes responses to my comments on the original manuscript from May 2019, but not my subsequent comments on the June 2020. For the most part, I think transferring the manuscript to Nature Communications is appropriate. Also, the authors have revised the manuscript to emphasize that the distinct role of ACC PT vs. IT neurons have not necessarily been explored in the context of addiction rather than making the overly broad and incorrect assertion that these have not been explored in behavior at all. As I wrote previously, "this is a solid study that will make a nice contribution to identifying circuits that mediate dissociable effects of cocaine on reward processing." As I wrote in my previous review, it does seem important to discuss a few ways in which these results relate to those of Meda et al., *Neuron*, 2019.

First, Meda et al. also looked at the role of ACC IT vs. PT neurons in reward vs. aversion using a CPP assay. I understand that Meda et al. focused on pain-related aversion whereas this study looked at cocaine-induced effects. However, both the title and abstract of this study focus on the idea that PT and IT cells in the ACC differentially contribute to reward and aversion. Thus, perhaps many readers would appreciate knowing about previous studies that have examined this general issue?

Second, Meda et al. also showed that MD thalamic inputs innervate PT cells – this seems important to mention since it illustrates the limitations of results obtained here with rabies tracing.

Third, Meda et al. also used the same method used here – CTb injection – to show that IT and PT populations are non-overlapping in layer 5 of ACC. The only real difference was that Meda et al. studied mouse, whereas this study is done in rats. Since this study basically finds the same thing, it seems appropriate to mention the prior study.

Reviewers' Comments:

Reviewer #1 (Remarks to the Author):

This is a provocative, exciting study that has potential to shift the field's thinking on the neural substrates underlying the rewarding and aversive effects of drugs of abuse. The rationale for the study and the significance of the findings are clearly articulated in most cases. The methods, in terms of targeting the IT/PT neuron populations are state-of-the-art, and the behaviors under study are well chosen. The manuscript has no significant weaknesses.

We thank the reviewer for their comments and previous critiques, which have improved our manuscript.

Reviewer #3 (Remarks to the Author):

The rebuttal document includes responses to my comments on the original manuscript from May 2019, but not my subsequent comments on the June 2020. For the most part, I think transferring the manuscript to Nature Communications is appropriate. Also, the authors have revised the manuscript to emphasize that the distinct role of ACC PT vs. IT neurons have not necessarily been explored in the context of addiction rather than making the overly broad and incorrect assertion that these have not been explored in behavior at all. As I wrote previously, "this is a solid study that will make a nice contribution to identifying circuits that mediate dissociable effects of cocaine on reward processing." As I wrote in my previous review, it does seem important to discuss a few ways in which these results relate to those of Meda et al., Neuron, 2019.

We thank the reviewer for their thoughts. We now include discussion of the Meda paper, as described in the point-by-point comments below and highlighted in yellow in the revised manuscript.

First, Meda et al. also looked at the role of ACC IT vs. PT neurons in reward vs. aversion using a CPP assay. I understand that Meda et al. focused on pain-related aversion whereas this study looked at cocaine-induced effects. However, both the title and abstract of this study focus on the idea that PT and IT cells in the ACC differentially contribute to reward and aversion. Thus, perhaps many readers would appreciate knowing about previous studies that have examined this general issue?

We now include chronic pain and reference to the Meda paper when talking about previous work of the anterior cingulate in the introduction:

For instance, the ACC is robustly activated by sexual mating and cocaine and -associated cues, as well as by chronic pain and remote fear memories^{5,6,7,8,9,10}.

We also include a discussion of the paper in the discussion:

Interestingly, studies in pain models of preference and aversion show that PT stimulation and inhibition have opposite effects on preference and aversion, which were input-dependent⁶, suggesting that the results observed here are unique to drug administration contexts.

Second, Meda et al. also showed that MD thalamic inputs innervate PT cells – this seems important to mention since it illustrates the limitations of results obtained here with rabies tracing.

We now include this information as well as discuss some differences in the two studies in our results section:

In contrast, inputs to IT neurons were much more diverse, including an especially strong input from the thalamus (Fig. 1E, 36% (113 of 315) of infected cells), consistent with other CTB tracing studies⁶. The relatively limited input we observed to PT neurons may be a result of the rabies tracing method or due to species differences, as previous work in mice demonstrated that ACC PT neurons are also innervated by the thalamus⁶.

Third, Meda et al. also used the same method used here – CTb injection – to show that IT and PT populations are non-overlapping in layer 5 of ACC. The only real difference was that Meda et al. studied mouse, whereas this study is done in rats. Since this study basically finds the same thing, it seems appropriate to mention the prior study.

We now highlight that the Meda paper also used CTb to show the populations were non-overlapping in the results section:

We found that 38% (33 of 86) of ACC neurons were labeled as putative PT (CTB-647+) neurons, 51.6% (45 of 86) were labeled as putative IT (CTB-488+) neurons, and 10.4% (8 of 86) were co-labeled with both fluorophores (Figure 2A,B); suggesting that ~90% of IT and PT neurons comprise distinct neuronal populations, consistent with previous CTB tracing studies⁶.